# Rényi Differential Privacy of Propose-Test-Release and Applications to Private and Robust Machine Learning

**Jiachen T. Wang**
Princeton University
tianhaowang@princeton.edu

**Saeed Mahloujifar**
Princeton University
sfar@princeton.edu

**Shouda Wang**
Princeton University
sw1041@princeton.edu

**Ruoxi Jia**
Virginia Tech
ruoxijia@vt.edu

**Prateek Mittal**
Princeton University
pmittal@princeton.edu

## Abstract

Propose-Test-Release (PTR) is a differential privacy framework that works with local sensitivity of functions, instead of their global sensitivity. This framework is typically used for releasing robust statistics such as median or trimmed mean in a differentially private manner. While PTR is a common framework introduced over a decade ago, using it in applications such as robust SGD where we need many adaptive robust queries is challenging. This is mainly due to the lack of Rényi Differential Privacy (RDP) analysis, an essential ingredient underlying the moments accountant approach for differentially private deep learning. In this work, we generalize the standard PTR and derive the first RDP bound for it when the target function has bounded global sensitivity. We show that our RDP bound for PTR yields tighter DP guarantees than the directly analyzed $(\varepsilon, \delta)$-DP. We also derive the algorithm-specific privacy amplification bound of PTR under subsampling. We show that our bound is much tighter than the general upper bound and close to the lower bound. Our RDP bounds enable tighter privacy loss calculation for the composition of many adaptive runs of PTR. As an application of our analysis, we show that PTR and our theoretical results can be used to design differentially private variants for byzantine robust training algorithms that use robust statistics for gradients aggregation. We conduct experiments on the settings of label, feature, and gradient corruption across different datasets and architectures. We show that PTR-based private and robust training algorithm significantly improves the utility compared with the baseline.

## 1   Introduction

Privacy is a major concern for deploying machine learning (ML). In response, differential privacy (DP) [DMNS06] has become the de-facto measure of privacy. For a differentially private mechanism, the probability distribution of the mechanism's outputs on a dataset should be close to the distribution of its outputs on the same dataset with any single individual's data replaced. A general recipe for releasing the value of a function $f$ on dataset $S$ in a differentially private way is adding random noise to $f(S)$ (output perturbation), where noise magnitude should scale with $f$'s global sensitivity.

However, it might be over-conservative to add noise scaled with global sensitivity. There is a line of research on whether we can do better (e.g., [NRS07, DL09, TS13, KNRS13]). Propose-Test-Release (PTR) [DL09] is a framework that improves the general recipe with the notion of *local sensitivity*.

36th Conference on Neural Information Processing Systems (NeurIPS 2022).

The main idea of PTR is as follows: instead of adding noise with respect to global sensitivity, we propose an amount of noise that is tolerable for many common queries. When we receive the actual query, we test (in a differentially private way) whether answering the query with the proposed amount of noise is enough for privacy. If the proposed noise is too small for limiting the privacy loss from the actual query, we may refuse to answer the query or respond with a larger noise. PTR works especially well for releasing robust statistics such as median or trimmed mean, as the robust statistics usually have small local sensitivity on most common inputs.

While PTR is a basic framework that ages back to the early days after the introduction of DP, it has not been used for differentially private optimization before. DP-SGD [ACG$^+$16] is the general backbone for differentially private deep learning and optimization. One major challenge of augmenting SGD with PTR and training DP models is the calculation of privacy parameters after a large number of adaptive compositions. Rényi differential privacy (RDP) and Moment Accountant [Mir17, ACG$^+$16] enable us to calculate tighter privacy parameters for training DP models. Without the RDP bound of PTR, we need to calculate the privacy parameters using the advanced composition theorems [DR$^+$14] that can lead to significantly looser privacy bounds. Besides, we also need the bound of *privacy amplification by subsampling* for PTR, which is the other important support for training DP models. It allows us to exploit the stochasticity of SGD for the interest of stronger privacy guarantees.

**Technical Overview.** In this work, we derive the Rényi DP bound for PTR, as well as for its Poisson subsampled variant when the target function has bounded global sensitivity. Our bounds make it possible for us to use PTR framework in augmenting private SGD. PTR could be characterized by three mechanisms. The first mechanism is $\mathcal{M}_1$ that determines which mechanism to run next. Depending on the outcome of $\mathcal{M}_1$ we then either run $\mathcal{M}_2$ or $\mathcal{M}'_2$. It is often the case that the worst-case privacy loss of one of $\mathcal{M}_2$ or $\mathcal{M}'_2$ is much larger than the other. Our bound exploits the fact that the worst-case will happen with small probability. Specifically, instead of considering the worst-case privacy loss between $\mathcal{M}_2$ and $\mathcal{M}'_2$, and naively composing it with $\mathcal{M}_1$, we show that for RDP we can tighten the bound by the *average* privacy loss of $\mathcal{M}_2$ and $\mathcal{M}'_2$ under the distribution imposed by $\mathcal{M}_1$. Direct $(\varepsilon, \delta)$-DP analysis does not enjoy this benefit; compared with directly analyzing the $(\varepsilon, \delta)$-DP bound of PTR, we show that by first bounding the RDP of PTR and then convert it to $(\varepsilon, \delta)$-DP can lead to better privacy guarantee. Our proof could serve as a general recipe for analyzing DP/RDP guarantees for composed mechanisms where the privacy loss of each mechanism is adaptively determined. Additionally, we extend our analysis to the RDP of *subsampled* PTR. Our algorithm-specific analysis (the "white-box bound") allows us to get tighter privacy amplification bounds, compared with the one obtained by general subsampled RDP bound that supports any mechanisms (the "black-box bound") [ZW19]. The proof tackled several additional difficulties compared with the analysis for simple Gaussian mechanism [MTZ19]. By numerical verification, we show that our RDP bound for subsampled PTR is much tighter than the black-box bound, and is close to the lower bound (Figure 2.).

**Applications of PTR in ML.** PTR is especially suitable for improving the utility of privatizing robust statistics such as trimmed mean, as these functions usually have a much smaller local sensitivity compared to their global sensitivity. A critical application of robust statistics for machine learning is defending against corrupted data. The learning algorithm should be robust in the presence of corrupted data (referred to as *Byzantine failure* [LSP82]). Since privacy and robustness are two major concerns for ML training, developing techniques that achieve both goals simultaneously is desirable. We demonstrate the application of PTR in incorporating differential privacy with robust SGD methods. A popular technique for robustifying SGD is to replace the mean with some robust statistics (e.g., trimmed mean) for gradient aggregation [YCKB18, AHJ$^+$21, GLV21]. We use trimmed mean as an example of showing how to augment robust SGD with DP through PTR. We show that the augmented SGD still maintains the robustness guarantee. We conduct extensive experiments on defending against three kinds of attacks: label, feature, and gradient corruptions, and we show that PTR-based robust SGD achieves much better utility than naively privatizing the robust SGD with global sensitivity.

## 2 Related Work

Since the seminal work of [DL09], the Propose-Test-Release has become a common DP framework mainly used in statistical inference [BAM20, LKO21]. To the best of our knowledge, this work is the first application of PTR in machine learning.

Several prior works (e.g., [LQS12, BBG18, WBK19, ZW19]) focus the general subsampled DP/RDP bound that supports any mechanisms. [WBK19] derives a general RDP privacy amplification for "sampling without replacement" scheme, and [ZW19] obtains a similar result under Poisson subsampling (i.e., including each data point independently at random with certain probability). Only a small body of recent works study algorithm-specific privacy amplification bound by subsampling. However, most of them focus on subsampled Gaussian mechanism [ACG+16, BDRS18, MTZ19]. This work derives the first subsampled RDP bound specific for flexible algorithms such as PTR.

Designing machine learning and optimization algorithms that achieve both privacy and Byzantine-robustness is certainly an important direction. Nevertheless, there are only few works on this line so far. [GGP+21] considered the problem of achieving privacy and byzantine resilience in distributed SGD with an untrusted server. The privacy level they considered is essentially local differential privacy, which is orthogonal to our focus. [HKJ20] and [SGA20] aim for both robustness and secure multiparty computation instead of differential privacy. [MZH19] naively applies differential privacy to defend against data poisoning attack. However, they find that DP alone cannot defend adversaries that poison a large fraction of training examples. Aside from the setting of training ML models, [LKKO21] propose a polynomial time algorithm that achieves both goals for mean estimation via privatizing filter-based robust mean estimator [DKK+17]. [EMN21] develop a robust and differentially private mean estimator based on exponential mechanism. However, both of their approaches become inefficient (even in polynomial time asymptotically) in high dimensional settings.

## 3 Background

In this section, we introduce some background on differential privacy, Rényi differential privacy, and privacy-amplification by subsampling. We will also introduce notations as we proceed.

**Differential Privacy.** Differential privacy is a framework for protecting privacy when performing statistical releases on a dataset with sensitive information about individuals (see the surveys [DR+14, Vad17]). Specifically, for a differentially private mechanism, the probability distribution of the mechanism's outputs of a dataset should be close to the distribution of its outputs on the same dataset with any single individual's data replaced. To formalize this, we call two datasets $S, S'$, each multisets over a data universe $\mathcal{X}$, *adjacent* if one can be obtained from the other by *adding or removing* a single element of $\mathcal{X}$. Further, we use $d(S, S')$ to denote the number of times of adding/removing of data points to transform $S$ to $S'$. So $S$ and $S'$ are adjacent if and only if $d(S, S') = 1$.

**Definition 3.1** (Differential Privacy [DMNS06]). For $\varepsilon, \delta \geq 0$, a randomized algorithm $\mathcal{M} :$ MultiSets$(\mathcal{X}) \to \mathcal{Y}$ is $(\varepsilon, \delta)$-*differentially private* if for every dataset pair $S, S' \in$ MultiSets$(\mathcal{X})$ such that $d(S, S') = 1$, we have:

$$\forall\, T \subseteq \mathcal{Y} \; \Pr[\mathcal{M}(S) \in T] \leq e^\varepsilon \cdot \Pr[\mathcal{M}(S') \in T] + \delta \tag{1}$$

where the randomness is over the coin flips of $\mathcal{M}$.

**Rényi Differential Privacy (RDP).** Rényi differential privacy (RDP) is a variant of the standard $(\varepsilon, \delta)$-DP that uses Rényi-divergence as a distance metric between the output distributions of $\mathcal{M}(S)$ and $\mathcal{M}(S')$, which is particularly useful in training differentially private machine learning models.

**Definition 3.2** (Rényi Differential Privacy [Mir17]). We say that a mechanism $\mathcal{M}$ is $(\alpha, \varepsilon)$-RDP with order $\alpha \in (1, \infty)$ if for every dataset pair $S, S' \in$ MultiSets$(\mathcal{X})$ such that $d(S, S') = 1$, we have:

$$D_\alpha \left( \mathcal{M}(S) \| \mathcal{M}(S') \right) := \frac{1}{\alpha - 1} \log \mathbb{E}_{o \sim \mathcal{M}(S')} \left[ \left( \frac{\mu_{\mathcal{M}(S)}(o)}{\mu_{\mathcal{M}(S')}(o)} \right)^\alpha \right] \leq \varepsilon \tag{2}$$

where $\mu_{\mathcal{M}}(\cdot)$ denotes the density function of $\mathcal{M}$'s distribution. Further, we denote the moment $E_\alpha \left( \mathcal{M}(S) \| \mathcal{M}(S') \right) := \mathbb{E}_{o \sim \mathcal{M}(S')} \left[ \left( \frac{\mu_{\mathcal{M}(S)}(o)}{\mu_{\mathcal{M}(S')}(o)} \right)^\alpha \right]$ and function $f_\alpha(\varepsilon) := \exp((\alpha - 1)\varepsilon)$.

As we can see, $(\alpha, \varepsilon)$-RDP is essentially an upper bound for the moment $E_\alpha \left( \mathcal{M}(S) \| \mathcal{M}(S') \right) \leq f_\alpha(\varepsilon)$ for all adjacent $S, S'$, where $\varepsilon$ can be viewed as a degree of the privacy loss incurred by running $\mathcal{M}$. A different $\alpha$ typically leads to a different privacy bound $\varepsilon$. Following the convention of literature [ZW19], we view $\varepsilon$ as a function of $\alpha$, and the notation $\varepsilon_{\mathcal{M}}(\alpha)$ means the algorithm $\mathcal{M}$ obeys

$(\alpha, \varepsilon_{\mathcal{M}}(\alpha))$-RDP. We note that RDP reduces to $(\varepsilon, 0)$-DP when we take $\alpha = \infty$. The current tightest $(\varepsilon, \delta)$-DP/RDP transformation we are aware of is by [ALC$^+$21]. A central property of DP/RDP is its behavior under composition. If we run multiple distinct differentially private algorithms on the same dataset, the resulting composed algorithm is also differentially private, with some degradation in the privacy parameters $(\varepsilon, \delta)$. Specifically, if we run $k$ sequentially chosen $(\varepsilon, \delta)$-DP algorithm on a dataset, the overall composed privacy parameter is $\left(\tilde{O}(\sqrt{k}\varepsilon), k\delta + \delta'\right)$-DP by Advanced composition theorem [DR$^+$14]. However, the Advanced composition theorem for $(\varepsilon, \delta)$-DP is loose. On the contrary, the composition is trivial for RDP as $\varepsilon_{\mathcal{M}_1 \circ \mathcal{M}_2}(\cdot) = \varepsilon_{\mathcal{M}_1}(\cdot) + \varepsilon_{\mathcal{M}_2}(\cdot)$. The Moment Accountant technique, which composes RDP and then transforms to DP, is a much simpler approach and often produces much more favorable privacy parameters than directly composing $(\varepsilon, \delta)$-DP. Therefore, RDP and Moment Accountant are widely used to calculate the privacy guarantee in training differentially private deep learning models.

**Privacy amplification by subsampling.** "Privacy amplification by subsampling" is the other booster besides RDP / moments accountant that drives much of the recent progress in differentially private deep learning. Most of the existing works focus on Poisson subsampling [BBG18], which samples each data point independently with a given sampling probability $q$. The tightest privacy amplification bound we are aware of for general mechanism under Poisson Subsampling is from [ZW19].

## 4  Propose-Test-Release and Rényi Differential Privacy

In this section, we introduce a general version of the Propose-Test-Release framework, present our main results on its RDP bound, and its privacy amplification bound under Poisson subsampling.

In our presentation, we use $\mathbb{GS}_f = \sup_{S,S':d(S,S')=1} \|f(S) - f(S')\|$ to denote the global sensitivity of target function $f$ (in $\ell_2$ distance), and we use $\mathbb{LS}_f(S) = \sup_{S,S':d(S,S')=1} \|f(S) - f(S')\|$ to denote the local sensitivity of function $f$ on dataset $S$. Before we present our main results, we would like to remind the readers about the definition of Laplace and Gaussian mechanisms, as well as their DP/RDP bound, which will be referred to later in the main results.

**Laplace Mechanism:** If $f$'s output is 1-dimensional, the Laplace mechanism $\mathcal{M}_{\mathrm{Lap},b}(S) = f(S) + \mathrm{Lap}(0, b)$ obeys $(\varepsilon_{\mathrm{Lap}}^{(\tilde{b})}, 0)$-DP for $\varepsilon_{\mathrm{Lap}}^{(\tilde{b})} = e^{1/\tilde{b}}$, and obeys $(\alpha, \varepsilon_{\mathrm{R-Lap}}^{(\tilde{b})}(\alpha))$-RDP for $\varepsilon_{\mathrm{R-Lap}}^{(\tilde{b})}(\alpha) = \frac{1}{\alpha-1} \log\left(\frac{\alpha}{2\alpha-1} \exp\left(\frac{\alpha-1}{\tilde{b}}\right) + \frac{\alpha-1}{2\alpha-1}\exp\left(-\frac{\alpha}{\tilde{b}}\right)\right)$, where $\tilde{b} = b/\mathbb{GS}_f$ (the "noise-to-sensitivity" ratio).

**Gaussian Mechanism:** If $f$'s output is $d$-dimensional, the Gaussian mechanism $\mathcal{M}_{\mathcal{N},\sigma}(S) = f(S) + \mathcal{N}(0, \sigma^2 \mathbf{1}_d)$ obeys $(\varepsilon_{\mathcal{N}}^{(\tilde{\sigma})}(\delta), \delta)$-DP for $\varepsilon_{\mathcal{N}}^{(\tilde{\sigma})}(\delta) = \tilde{\sigma}\sqrt{2\log(1.25/\delta)}$, and obeys $(\alpha, \varepsilon_{\mathrm{R-\mathcal{N}}}^{(\tilde{\sigma})}(\alpha))$-RDP for $\varepsilon_{\mathrm{R-\mathcal{N}}}^{(\tilde{\sigma})}(\alpha) = \frac{\alpha}{2\tilde{\sigma}^2}$, where $\tilde{\sigma} = \sigma/\mathbb{GS}_f$ (the "noise-to-sensitivity" ratio).

### 4.1  Propose-Test-Release

Naive use of Laplace/Gaussian mechanism may result in the poor utility of function output, as the global sensitivity of the target function may be intolerably large due to some extreme cases. Meanwhile, robust statistics such as median, mode, Inter-Quantile Range (IQR) are quite insensitive to single data addition/removal for datasets that are i.i.d. drawn from natural distributions. This means that robust statistics may have a small local sensitivity for most input datasets. The seminal work of [DL09] introduced Propose-Test-Release framework to reduce the noise addition when the target function has an approximation that is a robust statistic[1]. Here, we introduce a more general and useful version of PTR instantiated by Laplace and Gaussian mechanism. Given a target function $f_1$ (e.g., mean) and its robust variant $f_2$ (e.g., median), the PTR framework proceeds as follows: **(1) Propose:** a local sensitivity bound of the target function, $\tau$, is proposed; **(2) Test:** a safety margin $\Delta(S)$, which is the minimum amount of data points that we need to replace for $S$ to have local sensitivity larger than $\tau$, is computed. A private version (via Laplace mechanism) of the safety margin, $\widehat{\Delta}$, is compared with a threshold; **(3) Release:** if the safety margin is large enough, then the algorithm releases $f_2(S)$ via Gaussian mechanism with a smaller noise $\sigma_2$ (usually scaled with $\tau$). Otherwise, the algorithm

---

[1]Sometimes the robust statistic itself is the target function, i.e., $f_1 = f_2$ in Algorithm 1.

release $f_1(S)$ with a larger noise $\sigma_1$ (usually scaled with $\mathbb{GS}_{f_1}$). The pseudocode is outlined in Algorithm 1.

We make several remarks on Algorithm 1.

*Remark* 4.1. **(1)** The traditional version of PTR in the textbooks [DR$^+$14, Vad17] simply refuse to output anything when the noisy safety margin $\widehat{\Delta}$ is small. Algorithm 1 is a more general version which output $f_1(S) + \mathcal{N}(0, \sigma_1^2 \mathbf{1}_d)$ when the sensitivity test is failed. The textbook PTR can be thought of as a special case of Algorithm 1 where we set $\sigma_1 \to \infty$. **(2)** The mechanism used in the **Test** and **Release** step can be other mechanisms instead of Laplace and Gaussian. In this paper, we present our results for the PTR instantiated by these two mechanisms since we intend to apply it for differentially private SGD later. Our proof can be easily extended to other mechanisms as well. **(3)** The threshold $\log(1/(2\delta_0))b$

---

**Algorithm 1:** Propose-Test-Release with Laplace and Gaussian mechanism.

**input** : $S$ – dataset, $f_1$ – target function, $f_2$ – robust statistic, $\tau$ – proposed local sensitivity bound of $f_2$, $\sigma_1, \sigma_2, b$ – Gaussian/Laplace noise scales ($\sigma_1 > \sigma_2$), $\delta_0$ – failure probability.

1   $\Delta \leftarrow \min_{\tilde{S} \in \{\tilde{S}: \mathbb{LS}_{f_2}(\tilde{S}) > \tau\}} d\left(S, \tilde{S}\right)$.

2   $\widehat{\Delta} \leftarrow \Delta + \mathrm{Lap}(0, b)$.

3   **if** $\widehat{\Delta} \leq \log(1/(2\delta_0))b$ **then**

4     |   **return** $f_1(S) + \mathcal{N}(0, \sigma_1^2 \mathbf{1}_d)$

5   **else**

6     |   **return** $f_2(S) + \mathcal{N}(0, \sigma_2^2 \mathbf{1}_d)$

---

in Line 3 is chosen so that $\Pr[\mathrm{Lap}(0, b) > \log(1/(2\delta_0))b] = \delta_0$. **(4)** The global sensitivity of $\Delta(\cdot)$ is 1 for any functions. This could be seen by noticing that, for any pair of adjacent $S, S'$, we have $d(S, \tilde{S}) \leq d(S', \tilde{S}) + 1$ for any dataset $\tilde{S}$.

**Direct DP Analysis of Propose-Test-Release.** We show the DP guarantee for the Propose-Test-Release framework. To prove privacy, we need to find the worst pair of adjacent datasets $S$ and $S'$ that incurs the largest privacy loss. It is clear that the worst possible scenario of PTR is when $f_2(S) - f_2(S') > \tau$, while the algorithm still releases $f_2(S)$ with noise scaled with $\tau$. However, note that this worst possible scenario will only happen when *both* $\mathbb{LS}(S)$ *and* $\mathbb{LS}(S')$ *are greater than* $\tau$. In this case, however, $\Delta$ for both $S$ and $S'$ are 0 since there are no data points we need to change for them to have local sensitivity larger than $\tau$, and thus there is no privacy loss from $\widehat{\Delta}$. Moreover, when $\Delta = 0$, the probability that PTR will release $f_2(S) + \mathcal{N}(0, \sigma_2^2 \mathbf{1}_d)$ is at most $\delta_0$ by construction, which could be simply added to the $\delta$ term in $(\varepsilon, \delta)$-DP. If one of $\mathbb{LS}(S)$ and $\mathbb{LS}(S')$ is smaller than $\tau$, then we know that $f_2(S) - f_2(S') \leq \tau$, and the overall privacy parameters could be computed by Basic Composition Theorem [DR$^+$14]. We obtain the following differential privacy guarantee for PTR based on these observations. We defer more details of the proof to the Appendix A.

**Theorem 4.2** (Direct DP analysis for PTR). *Suppose* $\mathbb{GS}_{f_1} = \mathbb{GS}_{f_2} = 1$ *and* $\sigma_1 = \sigma_2/\tau$, *then Algorithm 1 is* $(\varepsilon_{\mathrm{Lap}}^{(b)} + \varepsilon_{\mathcal{N}}^{(\sigma_1)}(\delta), \delta_0 + \delta)$-*DP, where* $\varepsilon_{\mathrm{Lap}}^{(b)} = e^{1/b}$ *is the DP guarantee for Laplace mechanism, and* $\varepsilon_{\mathcal{N}}^{(\sigma_1)}(\delta) = \sigma_1 \sqrt{2 \log(1.25/\delta)}$.

## 4.2   RDP of Propose-Test-Release

We then derive the RDP of Propose-Test-Release framework. The major differences between the proofs of $(\varepsilon, \delta)$-DP and RDP bound is that, for $(\varepsilon, \delta)$-DP we can move the probability of running into the bad scenario to the $\delta$ term, while for RDP we need to consider the "average-case" privacy loss. This is in fact an advantage of RDP, which we illustrate it in the following simple example.

**Comparison between $(\varepsilon, \delta)$-DP and RDP Analysis: A motivating example.**[2] Suppose we have two mechanisms $\mathcal{M}_1$ and $\mathcal{M}_2$ who are $(\varepsilon_1, \delta_1)$-DP and $(\varepsilon_2, \delta_2)$-DP, respectively. Consider a simple PTR-like mechanism $\mathcal{M}$ that randomly picks one of mechanisms $\mathcal{M}_1$ and $\mathcal{M}_2$ to execute, each with probability $1 - \delta_0$ and $\delta_0$, respectively[3]. We can only claim that $\mathcal{M}$ is $(\max(\varepsilon_1, \varepsilon_2), (1 - \delta_0)\delta_1 + \delta_0\delta_2)$-DP, or if we know $\varepsilon_2 \gg \varepsilon_1$ we can also move the "bad case" probability $\delta_0$ to the $\delta$ term and obtain $(\varepsilon_1, \delta_0 + \delta_1)$-DP. However, if we know the RDP guarantee of $\mathcal{M}_1$ and $\mathcal{M}_2$ as $E_\alpha(\mathcal{M}_1(S) \| \mathcal{M}_1(S')) \leq f_\alpha(\varepsilon_1)$ and $E_\alpha(\mathcal{M}_2(S) \| \mathcal{M}_2(S')) \leq f_\alpha(\varepsilon_2)$[4], then $E_\alpha(\mathcal{M}(S) \| \mathcal{M}(S'))$

---

[2]Details of derivation can be found in Appendix A.

[3]For the actual PTR, the $\delta_0$ is not fixed but depends on the input dataset.

[4]Recall that $f_\alpha(\varepsilon) = \exp((\alpha - 1)\varepsilon)$ where if $\mathcal{M}$ is $(\alpha, \varepsilon)$-RDP then $E_\alpha(\mathcal{M}(S) \| \mathcal{M}(S')) \leq f_\alpha(\varepsilon)$.

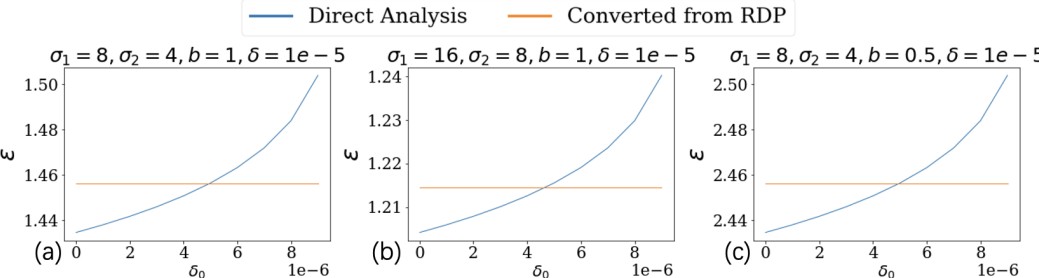

Figure 1: The $\varepsilon$ parameter of the $(\varepsilon, \delta)$-DP guarantee of PTR when $\delta = 10^{-5}$ for different noise scales. We convert the RDP bound in Theorem 4.3 to $(\varepsilon, \delta)$-DP by the RDP-DP conversion formula from [BBG+20], and compare it with the $\varepsilon$ obtained from the direct analysis in Theorem 4.2. For the bound converted from RDP, we search for the optimal $\alpha \in [1, 200]$. The bound is constant across different $\delta_0$ since when $\delta_0$ is small, the RDP for PTR will take the second term in (3).

can be simply bounded by $(1 - \delta_0)f_\alpha(\varepsilon_1) + \delta_0 f_\alpha(\varepsilon_2)$. Compared with $(\varepsilon, \delta)$-DP analysis, there are no extra inequalities used in RDP analysis of $\mathcal{M}$ except for the RDP guarantee of $\mathcal{M}_1$ and $\mathcal{M}_2$. Thus, RDP is more favorable in for PTR's privacy analysis, especially when $\delta_0$ is close to the target $\delta$.

**Theorem 4.3** (RDP analysis of PTR). *Suppose $\mathbb{GS}_{f_1} = \mathbb{GS}_{f_2} = 1$ and $\sigma_1 = \sigma_2/\tau$. Then for any $\alpha > 1$, Algorithm 1 is $(\alpha, \varepsilon_{\mathrm{PTR}}(\alpha))$-RDP for*

$$\varepsilon_{\mathrm{PTR}}(\alpha) \le \max\left(f_\alpha^{-1}\left((1 - \delta_0)f_\alpha\left(\varepsilon_{\mathrm{R-}\mathcal{N}}^{(\sigma_1)}(\alpha)\right) + \delta_0 f_\alpha\left(\varepsilon_{\mathrm{R-}\mathcal{N}}^{(\sigma_2)}(\alpha)\right)\right), \varepsilon_{\mathrm{R-}\mathcal{N}}^{(\sigma_1)}(\alpha) + \varepsilon_{\mathrm{R-Lap}}^{(b)}(\alpha)\right)$$

The above result implies that, with appropriately chosen $\delta_0$, the privacy loss of PTR is the same as directly adding up the privacy loss from $\widehat{\Delta}$ and from releasing $f_1(S) + \mathcal{N}(0, \sigma_1^2 \mathbf{1}_d)$ regardless of the value of $\widehat{\Delta}$. Compared with naively releasing $f_1(S) + \mathcal{N}(0, \sigma_1^2 \mathbf{1}_d)$, PTR will only pay an extra cost for privatizing $\Delta$, independent from the privacy cost for the case of releasing $f_2(S) + \mathcal{N}(0, \sigma_2^2 \mathbf{1}_d)$! Thus, the algorithm could potentially add much smaller noise while introducing just a little extra privacy loss. In Figure 1, we convert this RDP bound to $(\varepsilon, \delta)$-DP via the RDP-DP conversion formula by [BBG+20], and compare with the $(\varepsilon, \delta)$-DP bound obtained through direct analysis in Theorem 4.2. Although due to the loss in RDP-DP conversion, the $\varepsilon$ parameter by RDP is worse than the one by direct analysis when $\delta_0$ is far smaller than the target $\delta$, it is tighter when $\delta_0$ is close to $\delta$. A larger $\delta_0$ means PTR has a greater chance to release $f_2(S) + \mathcal{N}(0, \sigma_2^2 \mathbf{1}_d)$, which leads to a better utility. Thus, the privacy parameter converted from the RDP bound is more favorable.

We would like to stress that in the theorem, most of the conditions on the parameters such as $\mathbb{GS}$ or $\sigma_1, \sigma_2$ can be easily relaxed. The conditions in the theorems are mainly used to make the presented bound more clean and interpretable.

### 4.3 RDP for Poisson Subsampled Propose-Test-Release

We further derive the RDP for Poisson subsampled PTR. Here, "subsampled" means the input dataset will be subsampled first before feeding to PTR. Poisson subsampling means each data point $x$ will be included with probability $q$ independently. One way of obtaining the RDP for any subsampled mechanism is by simply plugging in the RDP bound of the original algorithm into the privacy amplification formula for general mechanisms (e.g., [WBK19, ZW19]). Here, we directly derive the RDP for Poisson subsampled PTR in a white-box manner. Denote $\mathcal{M}_0$ as the random variable for PTR's output on dataset $S$, and $\mathcal{M}_1$ for PTR's output on dataset $S' = S \cup \{x\}$, and $\mathcal{M} = (1 - q)\mathcal{M}_0 + q\mathcal{M}_1$. Compared with the analysis for subsampled Gaussian mechanism by [MTZ19], there are two main difficulties for extending the arguments of quasi-convexity of Rényi divergence for PTR: **(1)** the distribution of PTR's output may not be centrally symmetric, thus we need to bound both $D_\alpha(\mathcal{M}||\mathcal{M}_0)$ and $D_\alpha(\mathcal{M}_0||\mathcal{M})$. **(2)** the conditional distribution $\mathcal{M}|\widehat{\Delta}$ cannot be decomposed as $(1 - q)\mathcal{M}_0|\widehat{\Delta} + q\mathcal{M}_1|\widehat{\Delta}$. A big part of our novelty in the proof is about addressing these two challenges. We defer details of the proof to Appendix A.

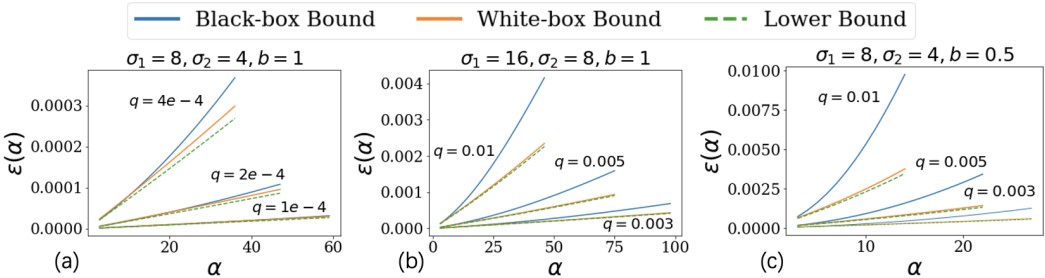

Figure 2: The RDP parameter $\varepsilon(\alpha)$ of the subsampled PTR as a function of order $\alpha$. We compare the white-box bound from Theorem 4.4, and the black-box bound as well as the lower bound by plugging in the RDP of PTR in Theorem 4.3 to the privacy amplification upper/lower bounds for general mechanisms from [ZW19].

**Theorem 4.4** (RDP analysis of sub-sampled PTR (the *"white-box bound"*)). *Let $q$ be the subsampling probability. Suppose $\mathbb{GS}_{f_1} = \mathbb{GS}_{f_2} = 1$ and $\sigma_1 = \sigma_2/\tau$. When $q, \sigma_1, \sigma_2$, and $\alpha$ satisfy certain conditions, we have*

$$\varepsilon_{\mathrm{PTR\circ PoissonSample}}(\alpha) \leq f_\alpha^{-1}\left(\max(\mathcal{B}_0, \mathcal{B}_1, \mathcal{B}_2)\right)$$

*where* $\mathcal{B}_0 = 1 + 2q^2\alpha(\alpha-1)(\frac{1-\delta_0}{\sigma_1^2} + \frac{\delta_0}{\sigma_2^2})$, $\mathcal{B}_1 = \mathrm{R}_q^{(\alpha)} + 2\alpha(\alpha-1)[\mathrm{R}_q^{(\alpha)} - 2(1-q)\mathrm{R}_q^{(\alpha-1)} + (1-q)^2\mathrm{R}_q^{(\alpha-2)}]$, $\mathcal{B}_2 = \widetilde{\mathrm{R}}_q^{(\alpha)} + 2\alpha(\alpha-1)[\widetilde{\mathrm{R}}_q^{(\alpha)} - 2(1-q)\widetilde{\mathrm{R}}_q^{(\alpha+1)} + (1-q)^2\widetilde{\mathrm{R}}_q^{(\alpha+2)}]$, *with* $\mathrm{R}_q^{(\alpha)} = \mathbb{E}_{s\sim\mu_0}\left[\left(\frac{\mu(s)}{\mu_0(s)}\right)^\alpha\right]$ *and* $\widetilde{\mathrm{R}}_q^{(\alpha)} = \mathbb{E}_{s\sim\mu}\left[\left(\frac{\mu_0(s)}{\mu(s)}\right)^\alpha\right]$ *for* $\mu_0 \sim \mathrm{Lap}(0,b)$ *and* $\mu \sim (1-q)\mathrm{Lap}(0,b) + q\mathrm{Lap}(1,b)$.

*Remark* 4.5. Both $\mathrm{R}_q^{(\alpha)}$ and $\widetilde{\mathrm{R}}_q^{(\alpha)}$ can be computed easily since they either have closed form or can be accurately computed via numerical integration in bounded range.

This bound may not be very interpretable, which is a typical feature for the privacy amplification bounds as they are meant to be implemented in practice. After all, constant matters for differential privacy practitioners! To show the tightness of our bound, we plug in the RDP of original PTR (Theorem 4.3) to the current tightest privacy amplification formula for general mechanisms derived in [ZW19] (the *"black-box bound"*), and compare it with our white-box bound in Theorem 4.4. As we can see from Figure 2, Theorem 4.4 (orange curve) is much tighter than the black-box bound (blue curve). Moreover, it is very close to the lower bound of privacy amplification by [ZW19] (green line), which means that Theorem 4.4 is near optimal. In Figure 3, we illustrate the application of subsampled RDP bound in Moment Accountant. We can see that the privacy parameters for the composed mechanism obtained based on our white-box bound (Theorem 4.4) is tighter than the one by black-box bound, as well as the one by directly composing Theorem 4.2 with strong composition theorem [KOV15]. We remark that Direct Analysis achieves lower privacy loss with very few iterations since we set $\delta_0 = 10^{-8}$ to allow more iterations for Strong Composition of $(\varepsilon, \delta)$-DP, which leads to a better $\varepsilon$ for a single iteration (see Figure 1).

An exciting byproduct during our research on subsampled PTR is the privacy amplification bound for a $\mathcal{M}$ that outputs $\mathcal{M}_1(S), \mathcal{M}_2(S)$ sequentially and *independently*. This result is important since such a $\mathcal{M}$ is instantiated in many existing techniques in improving differentially private deep learning. We defer this result to the Appendix.

## 5  Differentially Private and Robust SGD with Propose-Test-Release

In this section, we demonstrate the application of the RDP for (subsampled) PTR algorithm in privatizing robust SGD algorithms.

**Attack Model.** Mini-batch SGD is a common method for training deep neural networks. Despite its strong convergence properties in the standard settings, it is well known that even a small fraction of corrupted gradients can lead SGD to an arbitrarily poor solution [BTN00, BBC11]. An important attack model is called Byzantine contamination framework or *Byzantine failure* [LSP82]. Consider an

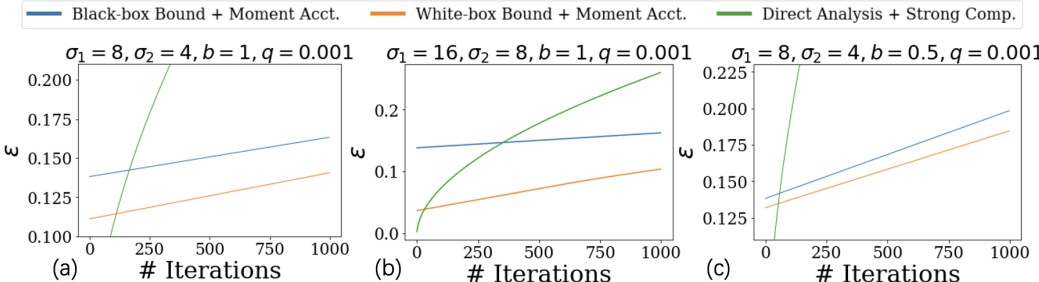

Figure 3: Illustration of the use of our Theorem 4.4 in moments accountant. We plot the the privacy loss $\varepsilon$ for $\delta = 10^{-5}$ after different rounds of composition. We set $\delta_0 = 10^{-8}$ here to allow more iterations for Strong Composition of $(\varepsilon, \delta)$-DP.

optimization problem with $n$ stochastic gradient oracles; at each iteration there are up to $F$ gradient oracles are corrupted (usually referred as *Byzantine agents* in the context of distributed SGD). The identity of corrupted oracles is a priori unknown. As the corrupted gradients can be arbitrarily skewed, this attack model is able to capture many important scenarios including **corruption in feature** (e.g., existence of outliers), **corruption in gradients** (e.g., hardware failure, unreliable communication channels during distributed training) and **corruption in labels** (e.g. label flip (backdoor) attacks).

A popular class of defense strategies against Byzantine failure is to use robust gradient aggregation rules such as trimmed-mean [GLV21], coordinate-wise median [YCKB18], and geometric median [AHJ+21]. Since both privacy and robustness are essential for ML applications, it is important to develop learning algorithms that achieve differential privacy and robustness simultaneously.

**Baseline: privatizing robust SGD with Gaussian Mechanism.** A natural (but naive) way to accomplish such a goal for SGD-based algorithms is to privatize each iteration of robust SGD through Gaussian mechanism. Specifically, in each iteration, we add Gaussian noise to the aggregated gradient, where noise magnitude scale with the global sensitivity of the robust aggregation function. The privacy parameter for the entire process is computed through Moment Accountant. The global sensitivity can be obtained by either upper bounding the Lipschitz constant of the objective function, or clipping each gradient according to a certain threshold. However, this method could potentially add over-conservative noise to the aggregated gradients, as the local sensitivity of robust aggregation functions is usually very low for most input gradient sets.

**Privatizing robust SGD with PTR.** We improve the naive method with the PTR framework, where we instantiate it by showing how to privatize trimmed-mean SGD method from [GLV21]. We first define trimmed-sum[5] aggregation function, and show that the $\Delta$ for trimmed-sum could be computed efficiently. Given a dataset $S = \{x_1, \ldots, x_m\}$, let $x_{(k)}$ denote the $k$th smallest data point among $S$ in $\ell_2$ norm, i.e., $\|x_{(1)}\| \leq \|x_{(2)}\| \leq \ldots \leq \|x_{(m)}\|$. The *F-trimmed sum of S* is defined as $\text{TSUM}_F(S) = \sum_{i=1}^{m-F} x_{(i)}$ if $m > F$, or simply 0 if $m \leq F$. Define $\mathbb{LS}_f^{(r)}(S) = \sup_{S,\tilde{S}:d(S,\tilde{S})=r} \mathbb{LS}_f(\tilde{S})$, which is the maximum local sensitivity that can be achieved by adding/removing $r$ elements from $S$. Thus, $\mathbb{LS}_f^{(0)}(S) = \mathbb{LS}_f(S)$, and the safety margin $\Delta(S) = \min\{r : \mathbb{LS}_f^{(r)}(S) > \tau\}$. Therefore, as long as we can efficiently compute $\mathbb{LS}_f^{(r)}(S)$, we can efficiently compute $\Delta(S)$ in linear time by simply enumerate all $r$ from 0 to $m$ and terminate once $\mathbb{LS}_f^{(r)}(S) > \tau$. In fact, $\mathbb{LS}_{\text{TSUM}_F}^{(r)}(S)$ can be computed in $O(1)$ time.

**Theorem 5.1.** $\mathbb{LS}_{\text{TSUM}_F}^{(r)}(S) = \|x_{(m-F+1+r)}\|$ *if* $r \leq F-1$, *or* $\mathbb{GS}_{\text{TSUM}_F}$ *if* $r > F-1$.

Back to the goal of designing differentially private and robust SGD. Fix a positive integer $F$ which is a potential upper bound for the number of corrupted gradients. We instantiate a PTR-based gradient aggregation algorithm by instantiating $f_1(\cdot)$ as $\text{SUM}(S) = \sum_{x \in S} x$, and $f_2(\cdot)$ as $\text{TSUM}_F(\cdot)$. We simply plugging in this PTR algorithm as the gradient aggregation function for regular SGD. We show

---

[5]We use trimmed-sum instead of trimmed-mean for the ease of analysis, as trimmed-mean essentially just scales the learning rate.

| Dataset | Corruption Type | CR | $\varepsilon = 3.0$ | | $\varepsilon = 5.0$ | |
|---|---|---|---|---|---|---|
| | | | TSGD+Gaussian | TSGD+PTR | TSGD+Gaussian | TSGD+PTR |
| **MNIST** | **Label** | **0** | 87.53% | 91.43% (+3.9%) | 81.31% | 92.8% (+11.49%) |
| | | **0.1** | 90.15% | 93.29% (+3.138%) | 89.15% | 94.608% (+5.456%) |
| | | **0.2** | 91.33% | 92.704% (+1.374%) | 92.47% | 94.084% (+1.614%) |
| | **Feature** | **0.1** | 89.47% | 92.28% (+2.812%) | 86.53% | 93.388% (+6.854%) |
| | | **0.2** | 91.71% | 92.29% (+0.582%) | 86.99% | 92.864% (+5.872%) |
| | **Gradient** | **0.1** | 90.27% | 91.7% (+1.43%) | 88.41% | 91.68% (+3.27%) |
| | | **0.2** | 91.13% | 91.45% (+0.32%) | 88.70% | 90.31% (+1.61%) |

| | | | $\varepsilon = 7.0$ | | $\varepsilon = 10.0$ | |
|---|---|---|---|---|---|---|
| | | | TSGD+Gaussian | TSGD+PTR | TSGD+Gaussian | TSGD+PTR |
| **CIFAR10** | **Label** | **0** | 57.94% | 58.92% (+0.98%) | 58.82% | 60.13% (+1.31%) |
| | | **0.1** | 55.48% | 56.262% (+0.78%) | 57.42% | 58.218% (+0.8%) |
| | | **0.2** | 49.69% | 50.946% (+1.26%) | 52.95% | 53.106% (+0.152%) |
| | **Feature** | **0.1** | 55.70% | 56.628% (+0.93%) | 57.36% | 58.678% (+1.318%) |
| | | **0.2** | 55.13% | 55.526% (+0.398%) | 56.93% | 57.336% (+0.404%) |
| | **Gradient** | **0.1** | 55.81% | 57.29% (+1.48%) | 57.21% | 58.96% (+1.75%) |
| | | **0.2** | 54.45% | 56.31% (+1.86%) | 56.11% | 57.48% (+1.37%) |

| | | | $\varepsilon = 2.0$ | | $\varepsilon = 2.5$ | |
|---|---|---|---|---|---|---|
| | | | TSGD+Gaussian | TSGD+PTR | TSGD+Gaussian | TSGD+PTR |
| **EMNIST** | **Label** | **0** | 72.94% | 76.44% (+3.5%) | 79.15% | 81.02% (+1.87%) |
| | | **0.1** | 72.60% | 75.66% (+3.06%) | 79.38% | 80.63% (+1.25%) |
| | | **0.2** | 70.03% | 72.62% (+2.59%) | 77.48% | 79.19% (+1.71%) |
| | **Feature** | **0.1** | 69.60% | 74.01% (+4.41%) | 77.80% | 81.04% (+3.24%) |
| | | **0.2** | 70.04% | 74.76% (+4.72%) | 78.99% | 80.74% (+1.75%) |
| | **Gradient** | **0.1** | 71.75% | 76.19% (+4.44%) | 77.32% | 77.73% (+0.41%) |
| | | **0.2** | 70.76% | 74.65% (+3.89%) | 76.68% | 77.17% (+0.49%) |

Table 1: Model Accuracy under different privacy budgets and corruption settings. Every statistic is averaged over 5 runs with different random seed. The improvement of $\mathrm{TSGD + PTR}$ over $\mathrm{TSGD + Gaussian}$ is highlighted in the red text. $\varepsilon$s are chosen differently for different datasets since the best accuracy-privacy tradeoff point varied for datasets. 'CR' means corruption ratio.

the convergence guarantee of this algorithm in the presence of at most $F$ gradients being corrupted for the case that the loss function is Lipschitz.

**Theorem 5.2** (Informal). *When the loss function is Lipschitz in model parameters, SGD with PTR instantiated by* SUM *and* TSUM *provides convergence guarantee for update to $F$ gradients being corrupted arbitrarily for any $F < m/2$.*

We make two remarks of our specific design choices for implementing PTR-based SGD in practice. *Remark* 5.3. **(1)** In practice, $F$ could be adjusted dynamically during the model training, based on the value of $\widehat{\Delta}$[6]. That is, if $\widehat{\Delta} \leq B$, it means that the current gradient batch has many outliers, thus we increase $F$ for the next iteration; if $\widehat{\Delta} > B$, it means that gradients are relatively concentrated and few of them are corrupted, thus we can decrease $F$ afterwards. **(2)** We set $f_1$ as SUM instead of TSUM since $F$ is usually set to be larger than the actual corrupted gradients; in this case, $\widehat{\Delta} \leq B$ means that there are many benign gradients also have large norms. These benign gradients usually correspond to the data points that are misclassified by the partially trained models, which is important for improving model performance. The pseudocode for the algorithm is deferred to the Appendix A.

## 5.1 Evaluation

In this section, we empirically evaluate the performance of PTR-based private and robust SGD. Specifically, we compare the utility-privacy tradeoff between the algorithm that privatize trimmed mean-based SGD with PTR ($\mathrm{TSGD + PTR}$), and the baseline algorithm that privatize trimmed mean-based SGD with Gaussian mechanism ($\mathrm{TSGD + Gaussian}$).

**Experiment Settings.** We evaluate the utility-privacy tradeoff of the two robust and differentially private SGD algorithms under three common types of attack: label, feature, and communicated gradient corruption. For label corruption, we randomly flip of label of certain amount of data points. For feature corruption, we add Gaussian noise from $\mathcal{N}(0, 100)$ directly to the corrupted images. For

---

[6]This does not introduce extra privacy leakage as DP is closed under post-processing.

gradient corruption, we add Gaussian noise from $\mathcal{N}(0, 100)$ to the true gradients. We experiment on three classic datasets, MNIST, CIFAR10, and EMNIST. We vary different corruption ratios and privacy parameter $\varepsilon$. Experiment details are deferred to Appendix.

**Results.** The comparison of model test accuracy under given corruption settings and privacy parameters are summarized in Table 1. We highlight the improvement of TSGD + PTR over the baseline TSGD + Gaussian in red texts. As we can see, for all settings, TSGD + PTR outperforms and often works significantly better than TSGD + Gaussian. This demonstrates that, while TSGD + PTR may introduce extra privacy loss in the **test** step, the performance gain from adding smaller noise in the **release** step overshadows it.

We also observe two interesting phenomena in the experiment: **(1)** a higher corruption ratio may not necessarily lead to worse model performance for trimmed mean-based robust SGD, especially for MNIST dataset. This is because the high norm gradients can be either corrupted, or benign but the partially-trained model misclassifies the corresponding data points. The latter case is extremely important for improving model performance compared with the gradients of data points are already being classified correctly. When the corruption ratio is small, more benign gradients are being trimmed, which may lead to worse model performance. **(2)** For TSGD + Gaussian, a larger privacy budget may not lead to better model performance on MNIST dataset. This is because when the training accuracy reaches the peak, there are many trimmed gradients whose corresponding training data points are already correctly classified; continuing training without those data points and with large noise may result in catastrophic forgetting.

# 6   Conclusion and Limitation

This work derives the Rényi Differential Privacy for propose-test-release framework as well as its subsampled version. With the RDP bound for the PTR framework, this work demonstrate the application of PTR in training differentially private and robust models. One limitation of PTR is that it does not work well in privatizing coordinate-wise median in high-dimensional space. The global sensitivity of coordinate-wise median is far greater than the one for mean, which results in huge privacy loss.

## Acknowledgement

We are grateful to anonymous reviewers at NeurIPS for valuable feedback. This work was supported in part by the National Science Foundation under grants CNS2131910, CNS-1553437, CNS-1553301, CNS-1704105, and CNS-1953786, the ARL's Army Artificial Intelligence Innovation Institute (A2I2), the Office of Naval Research Young Investigator Award, the Army Research Office Young Investigator Prize, Schmidt DataX award, Princeton E-ffiliates Award, Amazon-Virginia Tech Initiative in Efficient and Robust Machine Learning, and Princeton's Gordon Y. S. Wu Fellowship.

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
