# OpenReview forum: "Renyi Differential Privacy of Propose-Test-Release and Applications to Private and Robust Machine Learning"
_NeurIPS.cc/2022/Conference — NeurIPS 2022 Accept_

### Official Review · Reviewer_Tbda · 2022-07-10

**Rating:** 6
**Confidence:** 5
**Soundness:** 3 good
**Presentation:** 2 fair
**Contribution:** 2 fair

**Summary:**


Propose-test-release (PTR) is a general recipe for data-dependent DP analysis. PTR first proposes a local sensitivity bound and then tests the distance to the dataset such that the local sensitivity of which is larger than the predefined local sensitivity bound. The original PTR framework returns nothing when the test fails, while this work extends the PTR by

1.   modifying the PTR algorithm allows ``privacy for free'' when the safety margin is small.
2.  deriving the RDP bound for PTR approach as well as its subsampled variant.
3. applying PTR to the private SGD with robust estimates.

I have checked the appendix except the part of convergence analysis.

**Questions:**

1. In the proof of Theorem 4.3, Case 1 considers both the local sensitivity of $S$ and $S'$ are greater than $\tau$. Why requiring both of them instead of at least one of the local sensitivity being greater than $\tau$?  In my understanding, the private test shall be done for the dataset $S$ and $S'$ separately.

2. The authors claim that  the trimmed mean parameter $F$ could be adjusted dynamically during the model training. This part needs more clarification. If $F$ is a privacy parameter, then adaptively selecting $F$ requires the ``fully adaptive composition'' [1], rather than the general adaptive composition offered by the RDP accountant.

Small comments:
1. I was confused about the algorithm design on introducing two functions (target function f1 and the robust function f2) with the same global sensitivity. Usually, the PTR algorithm only deals with the robust function. However, the proposed PTR does not always release the output of f2. The extra output of releasing f1(S) might be unnecessary. If the authors replace f1 with f2, the privacy guarantee will still hold with a more precise logic.
2. The choice on $\sigma_1$ and $\sigma_2$ in two figures. The benefit of all data-dependent analysis is the assumption that ``the local sensitivity will be orders smaller than its global sensitivity counterpart''. However, the choices of $\sigma_1$ and $\sigma_2$ in the figures (and also the experiments) imply that the local sensitivity they consider is 0.5 while the global sensitivity is 1. I suggest the authors to try various choice of $\tau$ in the next version.


[1] Ryan M Rogers, Aaron Roth, Jonathan Ullman, and Salil Vadhan. Privacy odometers and filters:
pay-as-you-go composition

**Ethics Review Area:**

["I don’t know"]

**Limitations:**

Not applicable.

**Strengths And Weaknesses:**


The textbook PTR does not support the RDP-based analysis as they do not specify the global sensitivity of the target function. Once the global sensitivity is specified, the RDP version's extension is straightforward. However, the tighter RDP analysis of subsampled PTR is non-trivial as the proposed PTR does not satisfy the ``exact upper/lower RDP bound condition'' from the known literature. Moreover, the application to the private SGD with trimmed mean is novel.

---

> ### Author Response · Authors · 2022-08-02
> **Response to Reviewer Tbda**
>
> **Q1. [In the proof of Theorem 4.3, Case1, why to consider both the local sensitivity of S and S’ are greater than $\tau$ instead of analyzing S and S’ separately?]**
>
> **A:** DP (or RDP) are defined in terms of ALL PAIRS of neighboring datasets. Therefore, the proof logic is to consider an arbitrary pair of neighboring dataset $S$ and $S’$, and show that the distribution of $\mathcal{M}(S)$ and $\mathcal{M}(S’)$” are close in sup or Renyi divergence. To show that the divergence between $\mathcal{M}(S)$ and $\mathcal{M}(S’)$ are close, we separate it into two possible cases (1) the local sensitivity of both $S$ and $S’$ are greater than $\tau$, and (2) the local sensitivity of at least one of $S$ and $S’$ are smaller than $\tau$. The purpose of such separation is that, **for case (1), we know that the result of sensitivity test $\Delta(S)=\Delta(S’)=0$, which means that there is no privacy loss by releasing the result of $\widehat \Delta(S)$ or $\widehat \Delta(S’)$**. For case (2), we know that $|| f(S)-f(S’) || \le \tau$ as if the local sensitivity of $S$ is smaller than $\tau$, then $||f(S)-f(S’)|| \le \tau$ for all possible neighboring dataset $S’$. A similar argument applies if the local sensitivity of $S’$ is smaller than $\tau$. We have modified the proof to make the proof logic clear.
>
>
> **Q2. [Is $F$ a privacy parameter? Does dynamically adjusting $F$ affect privacy loss?]**
>
> **A:** $F$ is a parameter for trimmed mean function. The computation of the privacy bound in Theorem 4.3 and 4.4 does not require the information of $F$ (the global sensitivity of trimmed mean function is just the gradient clipping bound, which is independent of $F$), thus it is not a privacy parameter.
>
> Besides, $F$ is adjusted according to the value of $\widehat \Delta$ instead of $\Delta$. Therefore, $F$ is the post-processing of $\widehat \Delta$. Since DP/RDP are closed under post-processing, dynamically adjusting $F$ according to $\widehat \Delta$ does not introduce more privacy leakage. We have added a footnote on this point in the maintext.
>
> **Q3. [Why PTR algorithm in the paper separate target function $f_1$ and the robust function $f_2$?]**
>
> **A:** This design is mainly intended to keep the algorithm as general as possible. Indeed, PTR works with robust functions, and it is often the case that $f_1 = f_2$. However, consider the following scenario: one wants to estimate the mean of a dataset, which is not a robust statistic. Median is a good approximation for mean only when the dataset is concentrated and symmetric. The private sensitivity test can serve as a test for whether the input dataset satisfies such good concentration properties. If the dataset fails the sensitivity test, it means that the dataset may not be concentrated, and therefore median might not be a good approximation for the mean. In this case, releasing median and mean have the same privacy leakage but directly releasing the mean gives better utility.
>
> **Q4. [Experiment on more values of $\tau$.]**
>
> **A:** We sincerely thank the reviewer for the helpful suggestion. **We have included more experiment results in the Appendix B.5** with more choices of $\tau = \sigma_2 / \sigma_1$. As we can see, for all cases the privacy bounds derived by Theorem 4.3 and 4.4 are better than the corresponding baselines.

---

> ### Author Response · Authors · 2022-08-05
> **A Friendly Reminder**
>
> Dear Reviewer Tbda,
>
> We want to thank you for the positive comments about our paper. We’d also like to express our gratitude for your great questions, which led to a number of new revision improvements. We’ve responded to each of your questions. We are wondering if our response adequately addresses your concerns, and we would be delighted to provide further explanations within our allowed period.
>
> Best wishes,
>
> Authors of Paper

---

### Official Review · Reviewer_tPa6 · 2022-07-11

**Rating:** 6
**Confidence:** 3
**Soundness:** 3 good
**Presentation:** 4 excellent
**Contribution:** 3 good

**Summary:**

This paper applies Renyi DP to the propose-test-release (PTR) framework. The algorithmic difference between this work and the literature is that it changes "release nothing" by "releasing the result with a large noise". Then, armed with the RDP-based PTR, the authors improved the SGD in terms of differential privacy and robustness.

**Questions:**

1. Can the authors clarify the setting of robustness here?
2. Any results or intuition for the robustness of the proposed method?

**Limitations:**

I haven't found any negative social impact in this paper

**Strengths And Weaknesses:**

Strengths:
1. The framework is very interesting, and it considers two "big issues", robustness and privacy, at the same time
2. The PTR framework indeed provides an improvement in accuracy.

Weakness:
1. The definition of robustness is very unclear. I am not sure about which setting the authors studies. Adversarial attacks? Data poisoning? Or any other settings?
2. I didn't see any experimental results for the robustness.
3. In Table 1, CR is not defined. I think it means corruption rate.

---

> ### Author Response · Authors · 2022-08-02
> **Response to Reviewer tPa6**
>
> **Q1. [Setting of Robustness]**
>
> **A:** We have significantly expanded the discussion about attack models at the beginning of Section 5. Specifically, the attack model is called Byzantine contamination framework or *Byzantine failure* [1]. Consider an optimization problem with $n$ stochastic gradient oracles; at each iteration, up to $F$ gradient oracles are corrupted arbitrarily (usually referred to as *Byzantine agents* in the setting of distributed training). The identity of corrupted oracles is a priori unknown. By letting the corrupted estimates be arbitrarily skewed, this corruption model captures many important and practical scenarios including **feature corruption** (e.g., outliers in the data [2]), **gradient corruption** (e.g., malicious clients in federated learning [3, 4]), and **label corruption** (e.g., noisy label or backdoor attacks [5]).
>
> **Q2. [Results/intuition for the robustness of the proposed method?]**
>
> **A:** A popular way to defend against Byzantine failure is to replace mini-batch averaging with a robust gradient aggregation operator. The intuition is that by using a robust gradient aggregation function, the aggregated gradient will be affected less by corrupted individual gradients.
>
> As we clarified above, the Byzantine contamination framework captures many important scenarios such as **feature corruption**, **gradient corruption**, and **label corruption**. We simulate those possible corruptions in our experiment. We have added a detailed description of how we simulate those corruptions in Appendix B.1.1. Those corruption simulations experiments are standard in Byzantine robustness literature (e.g., see [3-7]).
>
> **During the rebuttal period, we additionally evaluate two more possible corruption types** that are considered more severe and adversarial:
> - **Gradient Bit-flipping failure** where the bits that control the sign of the floating numbers are flipped, e.g., due to some hardware failure. A faulty worker pushes the negative gradient instead of the true gradient to the servers.
> - **Targeted label flipping failure** where the labels are flipped in a "targeted" way. For example, for any *label* $\in {0, \ldots, 25}$, is replaced by $25-$ *label*. Such failures/attacks can be caused by data poisoning or software failures.
>
> The additional experiment results are shown in Appendix B.3. Once again, the robust SGD equipped with PTR outperforms the baseline technique, especially for the gradient bit-flipping attack (the improvement is at least 3%).
>
> [1] LAMPORT L, SHOSTAK R, PEASE M. The Byzantine Generals Problem. ACM Transactions on Programming Languages and Systems, 1982.
>
> [2] ​​Hendrycks D, Dietterich T. Benchmarking neural network robustness to common corruptions and perturbations. ICLR 2019.
>
> [3] Xie C, Koyejo S, Gupta I. Zeno: Distributed stochastic gradient descent with suspicion-based fault-tolerance. ICML 2019.
>
> [4] Bernstein J, Wang Y X, Azizzadenesheli K, et al. signSGD: Compressed optimisation for non-convex problems. ICML 2018.
>
> [5] Shen Y, Sanghavi S. Learning with bad training data via iterative trimmed loss minimization. ICML 2019.
>
> [6] Acharya A, Hashemi A, Jain P, et al. Robust training in high dimensions via block coordinate geometric median descent. AISTATS 2022.
>
> [7] Yin D, Chen Y, Kannan R, et al. Byzantine-robust distributed learning: Towards optimal statistical rates. ICML 2018.

---

> > ### Comment · Reviewer_tPa6 · 2022-08-08
> > **Thanks for the rebuttal**
> >
> > Thanks for the rebuttal and the revision! The rebuttal solved my concerns about the presentation. I have raised my score for "presentation", but will keep the over score.

---

> ### Author Response · Authors · 2022-08-05
> **A Friendly Reminder**
>
> Dear Reviewer tPa6,
>
> We want to thank you for the positive comments about our paper. We’d also like to express our gratitude for your great questions, which led to a number of new revision improvements. We’ve responded to each of your questions. We are wondering if our response adequately addresses your concerns, and we would be delighted to provide further explanations within our allowed period.
>
> Best wishes,
>
> Authors of Paper

---

### Official Review · Reviewer_6cEn · 2022-07-11

**Rating:** 6
**Confidence:** 3
**Soundness:** 3 good
**Presentation:** 3 good
**Contribution:** 3 good

**Summary:**

This paper derives an RDP bound for a generalized version of the Propose-Test-Release (PTR) framework, which is typically used to privately release robust statistics such as the median. The RDP analysis enables PTR to apply more readily to differentially private machine learning. The authors derive an RDP bound for Poisson-subsampled PTR, and use these tools to demonstrate how the PTR framework can improve the performance of robust SGD algorithms. Empirical results show that the PTR-based robust SGD offers an improvement over the baseline and is robust against corrupted data.


**Questions:**

1. Apart from trimmed mean-based SGD, are there other applications for PTR that would be effective for differentially private machine learning? And would any of these be outside the realm of robust statistics?

2. It seems like the experimental evaluation only compares trimmed mean-based robust SGD with and without PTR. How does trimmed mean-based robust SGD compare to its non-robust counterpart?


**Limitations:**

The authors adequately address the limitations of their work.

**Strengths And Weaknesses:**

The RDP analysis of PTR is novel and useful; it makes good progress in extending the PTR framework to machine learning applications. The paper is also nicely balanced with the empirical evaluation supporting the theoretical claims.

As for weaknesses, I do wonder about how widely applicable the PTR framework could be even with an RDP analysis. It seems like needing the privately released function to have small local sensitivity might be difficult to satisfy across all the variants of DP-SGD (especially the non-robust variants).

---

> ### Author Response · Authors · 2022-08-02
> **Response to Reviewer 6cEn**
>
> **Q1. [Other Applications of PTR?]**
>
> **A:** PTR is a paradigm for making robust statistics differentially private. As a general paradigm, PTR provides an advantage for privatizing any function whose local sensitivity is usually significantly smaller than global sensitivity. Moreover, **PTR-style mechanisms** (not necessarily the PTR), which first perform a private test and then execute different subroutines based on the test result, have many applications. For instance, PTR-style algorithms are used in PATE for prediction aggregation (see Algorithm 1 in [1]). The proof techniques in our paper can be used as a template for deriving the RDP guarantee for those flexible, data-dependent algorithms.
>
> [1] Papernot N, Song S, Mironov I, et al. Scalable private learning with pate, ICLR 2018.
>
>
> **Q2. [Are there any applications of PTR outside the realm of robust statistics?]**
>
> **A:** PTR is mainly designed for privatizing functions with local sensitivity significantly smaller than global sensitivity for most of the “common inputs”. For example, for regular DP-SGD, PTR may not provide extra advantages since mean usually does not have a small local sensitivity.
>
>
> **Q3. [How does trimmed mean-based robust and private SGD compare with regular DPSGD?]**
>
> **A:** We sincerely thank the reviewer for the constructive suggestion. **We include the additional experimental results on the comparison between trimmed mean-based robust SGD and regular DPSGD in Appendix B.4**. Overall, the robust SGD with PTR performs better than the non-robust counterpart in corrupted training data settings, especially for the setting of gradient corruption.

---

> ### Author Response · Authors · 2022-08-05
> **A Friendly Reminder**
>
> Dear Reviewer 6cEn,
>
> We want to thank you for the positive comments about our paper. We’d also like to express our gratitude for your great questions, which led to a number of new revision improvements. We’ve responded to each of your questions. We are wondering if our response adequately addresses your concerns, and we would be delighted to provide further explanations within our allowed period.
>
> Best wishes,
>
> Authors of Paper

---

### Official Review · Reviewer_ebzu · 2022-07-26

**Rating:** 4
**Confidence:** 3
**Soundness:** 3 good
**Presentation:** 3 good
**Contribution:** 2 fair

**Summary:**

The paper
1. shows the first RDP bound of Propose-Test-Release (PTR) algorithm in differential privacy.
2. shows another RDP bound of subsampled PTR
3. demonstrates that it is tighter in composition and applies it to robust SGD

**Questions:**

Does Figure show that your RDP bound can be worse than direct analysis? I don't quite understand why this could happen.

**Limitations:**

The main, and very personal concern is that the privacy of PTR is still not nailed. It's easily seen to have room to improve, for example, the algorithm doesn't actually output $\hat{\Delta}$ but the analysis assumes so. I don't have very involved experience with such analyses, but it looks to me that what needs to be considered is simply the divergences of two Gaussian mixtures, each with two components. Can't we obtain privacy parameters that truly respect the truth, so that no one has to worry about improving the analysis anymore?

I don't want to be too negative and discourage the authors, but I think less trotting and more big leaps could be more beneficial to the field.

**Strengths And Weaknesses:**

Picking up a classic work is great! I really appreciate the paragraph "Direct DP Analysis of Propose-Test-Release". The analysis is intuitive and clear. However, starting from section 4.2, the paper seems less readable to me compared to the early sections. For example, I didn't understand the $f_\alpha(\epsilon_1)$ and $f_\alpha(\epsilon_2)$ notations. It might be helpful to state the content of lines 221-223 as a lemma. Not surprisingly, I also don't understand the theorem for subsampled PTR.

---

> ### Author Response · Authors · 2022-08-02
> **Response to Reviewer ebzu**
>
> **Q1. [Motivating Example for the Comparison between DP and RDP analysis at the beginning of Section 4.2.]** *"However, starting from section 4.2, the paper seems less readable to me compared to the early sections. For example, I didn't understand the $f_\alpha(\epsilon_1)$ and $f_\alpha(\epsilon_2)$ notations. It might be helpful to state the content of lines 221-223 as a lemma."*
>
> **A:** Followed by the reviewer’s suggestion, we have modified the main text and condensed the content of lines 221-223 as an example paragraph. We also introduced some new notations to avoid complicated math expressions. For example, we additionally defined
> $E_\alpha \left(\mathcal{M}(S) ||  \mathcal{M}\left(S^{\prime}\right)\right)$
> in Definition 3.2. The definition of $f_\alpha(\epsilon) := \exp( (\alpha-1)\epsilon )$ can also be found in Definition 3.2, and we recall their definitions in the footnote of Section 4.2 for the convenience of readers. Furthermore, we show the detailed derivation of the motivating example in Appendix A.1 for the sake of completeness. We believe the motivating example is simple and intuitive, and we hope these changes can improve the readability of our paper.
>
>
>
> **Q2. [In Figure 1, why can RDP bound be worse than direct analysis?]**
>
> **A:** To compare the tightness between DP and RDP, we need to convert the RDP bound to $(\epsilon, \delta)$-DP via the RDP-DP conversion formula. This conversion is lossy (e.g., see the discussion in [1], Section 3). However, even with such a loss in RDP-DP conversion, the $\varepsilon$ parameter by RDP is still tighter than direct analysis when the $\delta_0$ parameter in RDP is close to the target $\delta$. An intuitive way to think about why RDP does not show advantages for the regime where $\delta_0$ is negligible is that, the smaller $\delta_0$, the closer the PTR to the regular mechanism, thus RDP can offer less advantage. Nevertheless, this loss will be compensated when we compose many mechanisms because the composition of RDP is tighter.
>
> [1] Zhu et al., Optimal Accounting of Differential Privacy via Characteristic Function, AISTATS 2022.
>
>
>
> **Q3. [Can we further improve the privacy analysis of PTR by not releasing $\widehat \Delta$?]** *"The main, and very personal concern is that the privacy of PTR is still not nailed. It's easily seen to have room to improve, for example, the algorithm doesn't actually output $\widehat \Delta$ but the analysis assumes so. I don't have very involved experience with such analyses, but it looks to me that what needs to be considered is simply the divergences of two Gaussian mixtures, each with two components. Can't we obtain privacy parameters that truly respect the truth, so that no one has to worry about improving the analysis anymore?"*
>
> **A:** *[tl;dr: Releasing $\widehat \Delta$ is important for user-friendly applications of PTR. ]*
>
> We thank the reviewer for the constructive suggestion. This is also one of the ideas we had at the beginning of this project. The main reason we decide to release $\widehat \Delta$ as the textbook PTR is that releasing $\widehat \Delta$ is essential for the applications of PTR. The rationale behind PTR is to exploit the fact that, while a function’s global sensitivity may be large, its local sensitivity may be much smaller for most of the “common inputs”. Thus, such a mechanism will only be preferred over a regular output perturbation mechanism when the local sensitivity of data drawn from input data distribution rarely exceeds the threshold. Without knowing about $\widehat \Delta$, the user **cannot** know whether they are actually enjoying the benefits from PTR or simply wasting privacy budgets on private sensitivity tests, as the results of these tests are unknown. Furthermore, the user cannot adjust the hyperparameters or switch algorithms accordingly. Notably, in Section 5 (the application of PTR in privatizing robust SGD), we also use the information from $\widehat \Delta$ to dynamically adjust the number of gradients to be trimmed (this does not affect privacy analysis since the adjustment is the post-processing of $\widehat \Delta$).
>
> **In Appendix A.1.1, we included the above discussion and also made an attempt to directly analyze the variant of PTR that does not release $\widehat \Delta$.** We do not see an easy way to obtain a better privacy bound than we have in Theorem 4.3.

---

> ### Author Response · Authors · 2022-08-05
> **A Friendly Reminder**
>
> Dear Reviewer ebzu,
>
> We’d like to express our gratitude for your constructive suggestions, which resulted in interesting revision updates. We’ve responded to each of your questions. We are wondering if our response adequately addresses your concerns, and we would be delighted to provide further explanations within our allowed period.
>
> Best wishes,
>
> Authors of Paper

---

### Author Response · Authors · 2022-08-02
**Summary of Changes to the Paper**

We thank all of the reviewers for the detailed and valuable comments. We are glad that our work receives a majority of positive reviews. We considered the reviews carefully and modified our paper accordingly. All modifications are highlighted. Here’s a summary of our major revision to the paper **(Appendix is directly followed by the references in the PDF file)**:
- **RDP of Propose-Test-Release (Section 4.2, Appendix A.1)**: we polished the writing of the motivating example to demonstrate why RDP may provide better privacy bounds for PTR compared with $(\varepsilon, \delta)$-DP. Furthermore, **we included the detailed derivation for this example in Appendix A.1**.
- **Discussion of why releasing $\widehat \Delta$ is important for the applications of PTR (Appendix A.1.1)**: we add a discussion explaining why $\widehat \Delta$ is important for the actual application of PTR.
- **Attack Model (Section 5, Appendix B.1.1)**: we expanded the description of our attack model (i.e., Byzantine contamination framework). **We also expanded the description of different Byzantine failure settings simulated in our experiment in Appendix B.1.1.**
- **Experiment with more Byzantine failure types (Appendix B.3)**: we experiment on two additional types of Byzantine failures: Gradient Bit-flipping failure and Targeted label flipping failure. These are two commonly studied corruption settings in Byzantine robustness literature (e.g., see [1-5]).
- **Comparison between trimmed mean robust SGD and regular DPSGD (Appendix B.4)**: we show the comparison between trimmed mean robust SGD and regular DPSGD in different corruption settings.
- **More numerical results for the Privacy Analysis Comparison between our Theorem 4.3/4.4 and baseline techniques (Appendix B.5)**: we show additional numerical results for the Privacy Analysis Comparison between our Theorem 4.3/4.4 and their corresponding baseline techniques by varying more hyperparameters.

[1] Xie C, Koyejo S, Gupta I. Zeno: Distributed stochastic gradient descent with suspicion-based fault-tolerance. ICML 2019

[2] Bernstein J, Wang Y X, Azizzadenesheli K, et al. signSGD: Compressed optimization for non-convex problems. ICML 2018.

[3] Shen Y, Sanghavi S. Learning with bad training data via iterative trimmed loss minimization. ICML 2019.

[4] Acharya A, Hashemi A, Jain P, et al. Robust training in high dimensions via block coordinate geometric median descent. AISTATS 2022.

[5] Yin D, Chen Y, Kannan R, et al. Byzantine-robust distributed learning: Towards optimal statistical rates. ICML 2018.

---

### Meta-Review · Area_Chair_FBjx · 2022-08-24

**Recommendation:** Accept
**Confidence:** Less certain

**Metareview:**

Paper studies RDP bound of Propose-Test-Release (PTR) algorithm in differential privacy.  In particular, it shows RDP bound of subsampled PTR and demonstrates how it is useful for composition of robust SGD. Given the textbook importance of PTR, we recommend accepting. However, we encourage authors to incorporate the comments from the authors, make sure that all the details of the proofs are made available in the final version, and clarify any comments reviewers raised.

**Award:**

No

---

### Decision · Program_Chairs · 2022-09-14

Accept